# Predictive factors requiring high-dose evocalcet in hemodialysis patients with secondary hyperparathyroidism

**Masanori Tokumoto**[1]*, **Shin Tokunaga**[2,3], **Shinji Asada**[2], **Yuichi Endo**[3], **Noriaki Kurita**[4,5,6], **Masafumi Fukagawa**[7], **Tadao Akizawa**[8]

**1** Department of Nephrology, Japanese Red Cross Fukuoka Hospital, Fukuoka, Japan, **2** Medical Affairs Department, Kyowa Kirin Co., Ltd., Tokyo, Japan, **3** R&D Division, Kyowa Kirin Co., Ltd., Tokyo, Japan, **4** Department of Clinical Epidemiology, Graduate School of Medicine, Fukushima Medical University, Fukushima, Japan, **5** Department of Innovative Research and Education for Clinicians and Trainees (DiRECT), Fukushima Medical University Hospital, Fukushima, Japan, **6** Center for Innovative Research for Communities and Clinical Excellence (CiRC2LE), Fukushima Medical University, Fukushima, Japan, **7** Division of Nephrology, Endocrinology, and Metabolism, Department of Internal Medicine, Tokai University School of Medicine, Kanagawa, Japan, **8** Division of Nephrology, Department of Medicine, Showa University School of Medicine, Tokyo, Japan

* m-tokumoto@fukuoka-med.jrc.or.jp

**Data Availability Statement:** All relevant data are within the paper and its Supporting Information files.

## Abstract

The dosage of evocalcet required to control serum parathyroid hormone (PTH) levels varies among secondary hyperparathyroidism (SHPT) patients. This post hoc analysis evaluated the dose-dependent efficacy of evocalcet on serum intact PTH (iPTH) levels, corrected calcium (Ca) and phosphate (P) levels, and safety, in an evaluation period (week 28 to week 30) by stratifying the previous phase 3 data with the final evocalcet dosages (low 1–2 mg [131 patients], medium 3–4 mg [90 patients], high 5–8 mg [92 patients]), and identified pre-treatment patient characteristics predicting the use of higher final evocalcet dosages via uni-variate and multivariate logistic regression models. At the end of the study at week 30, the median serum iPTH level was higher and the achievement ratio for the target range of Japanese Society for Dialysis Therapy (60–240 pg/mL) was lower in the final high-dose subgroup (216 pg/mL and 58%, respectively) than in the other subgroups (low: 149 pg/mL and 79%; medium: 149 pg/mL and 73%, respectively). Among the three subgroups, the mean serum corrected Ca and P levels demonstrated similar trends, and similar ratio of patients achieved the target range (corrected Ca, 8.4–10 mg/dL; P, 3.5–6.0 mg/dL) from week 28 to week 30. No dose-dependent safety concerns were identified. Younger age, prior cinacalcet use, higher serum levels of iPTH and corrected Ca, procollagen type 1 N-terminal propeptide, intact fibroblast growth factor-23, and larger maximum parathyroid gland volume were significantly associated with final high-dose evocalcet ($p < 0.05$ in all cases). Patients requiring final high-dose evocalcet had pre-treatment characteristics indicating severe SHPT, leading to a lower final achievement rate for the target PTH levels of Japanese Society for Dialysis Therapy. Therefore, the early initiation of evocalcet treatment for SHPT is critical.

**Trial registration:** This trial was registered as follows: ClinicalTrials.gov: NCT02549391 and JAPIC: JapicCTI-153013.

**Funding:** The study was funded by Kyowa Kirin Co., Ltd. The funder provided support in the form of consultancy fees and lecture fees to authors MT, MF and TA, salaries to authors ST, SA, and YE, lecture fees and travel grants (not connected to this research) to author NK, and grant to author MF. The funder was involved in the study design, data collection and analysis, decision to publish, and preparation of manuscript.

**Competing interests:** MT received consulting fees from KKC and Ono Pharmaceutical, and lecture fees from KKC, Chugai Pharmaceutical, Bayer, Kissei Pharmaceutical, Torii Pharmaceutical, Fuso Pharmaceutical, and Ono Pharmaceutical. ST, SA, and YE are employees of KKC. NK received lecture fees and travel grants for research meetings from KKC, which are not connected to this research. MF received consulting fees from KKC and Ono Pharmaceutical, lecture fees from KKC, Bayer, Torii Pharmaceutical and Ono Pharmaceutical, and grants from KKC and Bayer. TA received consulting fees from KKC, Astellas Pharma, Bayer, Fuso Pharmaceutical, Japan Tobacco, Ono Pharmaceutical, Sanwa Chemical, Otsuka, GSK, and NIPRO, and lecture fees from KKC, Chugai Pharmaceutical, Bayer, Kissei Pharmaceutical, Torii Pharmaceutical, and Ono Pharmaceutical. This does not alter our adherence to PLOS ONE policies on sharing data and materials.

## Introduction

Secondary hyperparathyroidism (SHPT) is a major complication in patients with chronic kidney disease [1–4]. Elevated serum levels of parathyroid hormone (PTH), calcium (Ca), and phosphate (P) in SHPT patients are associated with an increased risk of cardiovascular morbidity, as well as all-cause and cardiovascular-related mortality [5–7]. Therefore, it is critical to adequately control SHPT.

Allosteric modulators of the Ca-sensing receptor, such as cinacalcet hydrochloride (cinacalcet), can reduce serum levels of PTH, Ca, and P in SHPT patients [8]. Evocalcet is a new oral calcimimetic developed to decrease the gastrointestinal adverse events observed with cinacalcet [9–16]. A daily dose ranging from 1 to 8 mg of evocalcet is administered orally by combining 1 and 2 mg tablets, and the dose can be increased up to 12 mg in accordance with the approved dose range if the efficacy is insufficient. In a previous phase 3 long-term study of evocalcet, SHPT was controlled at a low dose in some patients, although a higher dose was required in other patients [17]. Because the expression of Ca-sensing receptors is decreased in the enlarged parathyroid glands (PTGs) of patients with severe SHPT [18], it is thought that SHPT may be advanced in patients who require a high dose of calcimimetics.

It would be beneficial for physicians to know which factors are associated with patients who are likely to require final high evocalcet doses to control serum levels of PTH, Ca, and P. Therefore, the present study identified a set of pre-treatment characteristics in hemodialysis patients with SHPT who required higher final evocalcet dose ranges, and evaluated the efficacy and safety of evocalcet in patients treated with final low-, medium-, and high-dose ranges by analyzing the data of the previous phase 3 head-to-head comparison study (the parent study) [12].

## Materials and methods

This study was conducted in accordance with the principles of the Declaration of Helsinki, the Pharmaceuticals, Medical Devices, Other Therapeutic Products Act, Good Clinical Practice (Ministry of Health and Welfare Ordinance No. 28, dated March 27, 1997), and the partial revision of the Ordinance after it was approved by the ethics committee at each study site (89 sites; S1 Table). Before the study started, signed informed consent was obtained from each patient.

### Parent phase 3 head-to-head comparison study

The parent randomized, double-blind, intrapatient dose-adjustment, parallel-group design phase 3 study using cinacalcet as an active control was conducted in hemodialysis patients with SHPT to evaluate the efficacy and safety of 30 weeks of treatment (28 weeks of dose adjustment period followed by a 2-week evaluation period) with once-daily oral evocalcet at 89 sites in Japan from October 2015 to November 2016 [12]. During the dose adjustment period, treatment with evocalcet was initiated at 1 or 2 mg in evocalcet-grouped patients with serum iPTH levels <500 or ≥500 pg/mL at screening, respectively. The dose was then adjusted at 1 mg increments up to 8 mg on the day of dialysis to control the serum iPTH level within the Japanese Society for Dialysis Therapy (JSDT) target range of 60–240 pg/mL [19] if an ongoing treatment dose was administered for ≥3 weeks, serum iPTH level was >240 pg/mL, and serum corrected Ca level was ≥8.4 mg/dL at the last visit before a dose change. Dose escalation was prohibited during the evaluation period. The dose was reduced if patients had a serum iPTH level <60 pg/mL, or investigators determined that a dose reduction was needed because of adverse events. The administration was interrupted if patients had a serum corrected Ca level ≤7.5 mg/dL, or investigators determined that a discontinuance was required because of

adverse events. An alternation of the prescribed dialysis conditions, including the Ca concentration in the dialysate and vitamin D receptor activator (VDRA) medications from 2 weeks before the screening until week 30 (or at the time of study discontinuation) was prohibited. Investigators were instead allowed to increase or initiate the Ca preparation as needed. Concomitant use of cinacalcet, bisphosphonates, teriparatide, and denosumab, parathyroid intervention, including parathyroidectomy, and peritoneal dialysis were prohibited during the entire study period.

Patients included in the parent study were ≥20 years old and had been undergoing hemodialysis three times a week for ≥12 weeks before screening. The mean serum iPTH level immediately before the start of study treatment was >240 pg/mL.

Serum levels of iPTH, corrected Ca, and P were measured at screening and every week from week 0 to week 30. The administrated dosages of evocalcet and cinacalcet were examined from week 0 to week 30, and concomitant medications were monitored from screening to week 30. Adverse events were monitored from week 0 to week 30.

The mean (standard deviation [SD]) of the medication adherence rate for evocalcet was 98.51% (6.24%) and that for cinacalcet was 98.91% (2.24%).

## Study design

This post hoc analysis included patients who were treated with evocalcet in the parent study (ClinicalTrials.gov, NCT02549391 and JAPIC, JapicCTI-153013). This study was designed to evaluate the efficacy and safety of evocalcet in the evaluation period from week 28 to week 30 according to the evocalcet final dose range administered to patients at week 28, and to identify week 0 characteristics of patients who required the medium- and/or high-dose evocalcet ranges in week 28. Therefore, data were analyzed by stratifying patients according to three ranges of daily evocalcet dose at week 28 after final dose adjustment: low-dose 1–2 mg (1 tablet), medium-dose 3–4 mg (2 tablets), and high-dose 5–8 mg (3–4 tablets) ranges. If the evocalcet dose data at week 28 were missing because the medication was interrupted or a patient discontinued the study, the dose prescribed right before week 28 was used.

## Evaluation of efficacy and safety

In this study, the serum levels of iPTH, corrected Ca, and P, and the ratio of patients who achieved these target ranges of JSDT guideline from week 28 to week 30, according to the evocalcet dose range administered to patients after final dose adjustment, were evaluated for the efficacy analyses. For the analysis of safety, adverse drug reactions (ADRs), especially hypocalcemia-related ADRs (corrected Ca decreased, blood Ca decreased, and hypocalcemia), and gastrointestinal tract-related ADRs (nausea, vomiting, abdominal discomfort, abdominal distension, and decreased appetite) from week 28 to week 30, according to the three ranges of daily evocalcet dose after final dose adjustment, were evaluated.

## Analysis for pre-treatment predictive factors associated with final evocalcet dosages

Patient characteristics at week 0 associated with medium and/or high final evocalcet dose ranges were retrospectively identified by a logistic regression model.

## Statistical methods

Efficacy endpoints were evaluated in the full analysis set, which included all enrolled patients, except those who did not receive evocalcet or had no iPTH measurements after the initiation

of treatment with evocalcet. Safety endpoints were evaluated in the safety analysis set, in which all patients who received evocalcet were included. Pre-treatment predictive factors associated with final evocalcet dosages were evaluated in the full analysis set.

The patient characteristics evaluated at week 0 (for the identification of predictive factors) and week 28 (for the efficacy and safety analyses) in this study were sex, age, primary kidney disease, duration of dialysis, type of dialysis, dry weight, dialysis efficiency, prior use of cinacalcet, use of intravenous VDRA and P binders, serum levels of iPTH, corrected Ca, P, bone-specific alkaline phosphatase (BAP), tartrate-resistant acid phosphatase 5b (TRACP-5b), procollagen type 1 *N*-terminal propeptide (P1NP), and intact fibroblast growth factor 23 (iFGF23), and the maximum PTG volume. Frequencies and ratios were obtained for nominal and rank data at each dose range. The mean and SD, or median and interquartile range, were calculated for continuous data in all patients as well as in each dose subgroup. The statistical significance of differences in dose subgroups was estimated using the $\chi^2$ test (two-tailed) for categorical variables and the Kruskal–Wallis test and analysis of variance (two-tailed) according to the distribution for continuous variables.

For the efficacy endpoints, the median (interquartile range) was obtained for serum iPTH levels, and the mean (SD) was calculated for corrected Ca and P levels from week 28 to week 30 in each subgroup. The ratio of patients who achieved target levels of JSDT for iPTH (60–240 pg/mL), corrected Ca (8.4–10.0 mg/dL), and P (3.5–6.0 mg/dL) [19] were calculated from week 28 to week 30 using those who had levels above, within, and below the target ranges. The serum levels of iPTH, corrected Ca, and P, and the ratio of patients who achieved the target ranges of JSDT guidelines from week 0 to week 28, according to the evocalcet dose range administered to patients at week 28 after final dose adjustment, were also presented as references.

For the safety analysis, all ADRs from week 28 to week 30 were categorized using MedDRA version 19.0, and incidence rates were calculated for all ADRs, hypocalcemia-related ADRs, and gastrointestinal tract-related ADRs in each subgroup. The same analysis was also applied to all ADRs from week 0 to week 28 in each subgroup, as references.

To identify pre-treatment predictive factors, a logistic regression model was applied to assess the association between patient characteristics at week 0 and the likelihood of being in the medium- and/or high-dose ranges at week 28, compared with the low-dose range, which was treated as a base outcome. The following variables were used: sex, age, primary kidney disease, duration of dialysis, type of dialysis, dialysis efficiency, prior use of cinacalcet, use of intravenous VDRA and P binders, serum levels of iPTH, corrected Ca, P, P1NP, and iFGF23, and the maximum PTG volume.

SAS version 9.4 (SAS Institute, Cary, NC, USA) was used for statistical analyses, and $p < 0.05$ was considered statistically significant.

## Results

In the parent study, 942 patients were screened for eligibility, and 639 patients were randomized to receive either evocalcet (320 patients) or cinacalcet (319 patients) (Fig 1) [12]. Of 320 patients randomized to receive evocalcet, 317 patients were included in the safety analysis set because 3 patients did not receive evocalcet. In the full analysis set, 313 patients were included.

### Efficacy

At week 0, the mean (SD) age of patients was 61 (11) years, 221 patients (71%) were male, and 85 patients (27%) had diabetic nephropathy. The mean (SD) duration of dialysis was 130 (89) months, and 247 (79%) and 66 (21%) patients were undergoing hemodialysis and

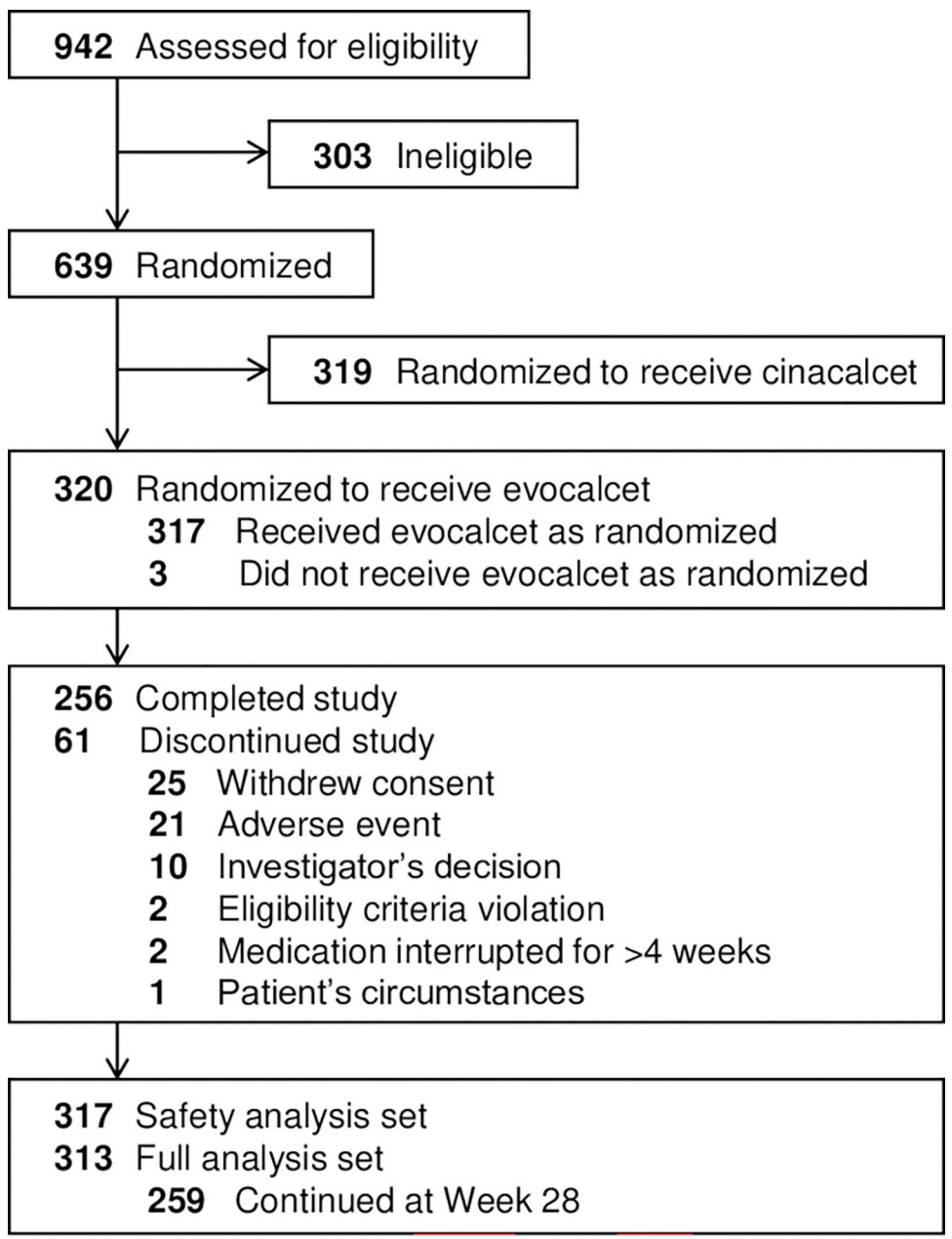

**Fig 1. Patient flow diagram.**

hemodiafiltration, respectively. The mean (SD) dialysate Ca concentration at week 0 was 2.78 (0.20). Overall, 59% of patients were on cinacalcet before screening and 52% were receiving intravenous VDRA at week 0 (Table 1).

The analysis of efficacy showed that median (interquartile range) serum iPTH levels were stable during the evaluation period with higher levels in the high-dose subgroup, with 149 (99, 212), 149 (108, 197), and 216 (136, 341) pg/mL in the low-, medium-, and high-dose subgroups, respectively, at week 30 (Fig 2A, S1 File). Mean (SD) serum corrected Ca levels in the three subgroups remained at similar levels in the evaluation period until week 30 (8.8 [0.7], 8.7 [0.7], and 8.7 [0.6] mg/dL, respectively; Fig 2B, S1 File). Mean serum P levels in the three

**Table 1. Patient characteristics at week 0 in each subgroup stratified by the final evocalcet dosages.**

| Parameter | | ALL | Final evocalcet dosages (mg/day) | | | p-value |
|---|---|---|---|---|---|---|
| | | | 1–2 | 3–4 | 5–8 | |
| | | n = 313 | n = 131 | n = 90 | n = 92 | |
| Sex, n (%) | Male | 221 (71) | 92 (70) | 66 (73) | 63 (69) | 0.77 |
| Age, years | | 61 ± 11 | 65 ± 11 | 59 ± 11 | 58 ± 11 | <0.001 |
| Height, cm | | 163 ± 8 | 161 ± 8 | 165 ± 8 | 164 ± 9 | 0.003 |
| Dry weight, kg | | 62 ± 14 | 61 ± 12 | 64 ± 16 | 63 ± 13 | 0.15 |
| Primary kidney disease, n (%) | DN | 85 (27) | 47 (36) | 21 (23) | 17 (19) | <0.001 |
| | CGN | 130 (42) | 39 (30) | 37 (41) | 54 (59) | |
| | NS | 37 (12) | 17 (13) | 7 (8) | 13 (14) | |
| | Others | 61 (20) | 28 (21) | 25 (28) | 8 (9) | |
| Duration of dialysis, months | | 130 ± 89 | 104 ± 82 | 136 ± 91 | 160 ± 86 | <0.001 |
| Type of dialysis, n (%) | HD | 247 (79) | 112 (86) | 72 (80) | 63 (69) | 0.009 |
| | HDF | 66 (21) | 19 (15) | 18 (20) | 29 (32) | |
| Dialysis efficiency (spKt/V) | | 1.49 ± 0.29 | 1.46 ± 0.28 | 1.50 ± 0.28 | 1.52 ± 0.32 | 0.34 |
| Dialysate Ca concentration, mEq/L | | 2.78 ± 0.20 | 2.78 ± 0.19 | 2.77 ± 0.21 | 2.79 ± 0.20 | 0.64 |
| Prior cinacalcet use, n (%) | Yes | 186 (59) | 51 (39) | 57 (63) | 78 (85) | <0.001 |
| Intravenous VDRA use, n (%) | Yes | 164 (52) | 54 (41) | 52 (58) | 58 (63) | 0.003 |
| P binder use, n (%) | Yes | 291 (93) | 118 (90) | 85 (94) | 88 (96) | 0.22 |
| Serum iPTH level, pg/mL, | | 374 (298, 460) | 311 (272, 393) | 382 (314, 458) | 458 (367, 596) | <0.001 |
| Serum corrected Ca level, mg/dL | | 9.5 ± 0.6 | 9.3 ± 0.5 | 9.5 ± 0.5 | 9.8 ± 0.6 | <0.001 |
| Serum P level, mg/dL, | | 5.8 ± 1.3 | 5.7 ± 1.3 | 5.9 ± 1.3 | 5.8 ± 1.4 | 0.59 |
| Serum BAP level, μg/L | | 18 ± 10 | 17 ± 10 | 18 ± 11 | 18 ± 9 | 0.30 |
| Serum TRACP-5b level, mU/dL | | 785 ± 406 | 723 ± 414 | 736 ± 358 | 920 ± 409 | <0.001 |
| Serum P1NP level, μg/L | | 358 (221, 544) | 300 (185, 443) | 347 (188, 536) | 434 (317, 712) | <0.001 |
| Serum iFGF23 level, pg/mL | | 13,300 (5670, 28,200) | 11,000 (3040, 23,200) | 13,900 (6310, 25,400) | 21,300 (7800, 39,900) | <0.001 |
| Maximum PTG volume, mm$^3$ | | 311 ± 574 | 211 ± 504 | 278 ± 340 | 455 ± 769 | <0.001 |

Data are expressed as the mean (standard deviation), median (interquartile range), or number (percentage). The statistical significance of differences for categorical variables was estimated using the $\chi^2$ test, and continuous variables were assessed using the Kruskal–Wallis test and analysis of variance.

BAP, bone-specific alkaline phosphatase; Ca, calcium; CGN, chronic glomerulonephritis; DN, diabetic nephropathy; HD, hemodialysis; HDF, hemodiafiltration; iFGF23, intact fibroblast growth factor 23; iPTH, intact parathyroid hormone; NS, nephrosclerosis; P, phosphate; P1NP, procollagen type 1 N-terminal propeptide; PTG, parathyroid gland; TRACP-5b, tartrate-resistant acid phosphatase-5b; VDRA, vitamin D receptor activator.

subgroups remained at the same levels during the evaluation period (approximately 5 mg/dL) (Fig 2C, S1 File). Median (interquartile range) serum iPTH levels gradually decreased during the dose adjustment period. Mean (SD) serum corrected Ca levels in the three subgroups decreased to similar levels by week 10 and remained at similar levels over the rest of the dose adjustment period. Mean serum P levels in the three subgroups were similarly decreased to below the respective week 0 levels and remained at the same levels over the rest of the dose adjustment period.

The ratio of patients who achieved the target iPTH level of JSDT in the final low-dose subgroup subsequently reached approximately 80% (Fig 3A, S2 File) and was similar in the final medium-dose subgroup, but it was lower in the high-dose subgroup at week 30 (medium 73% and high 58%). The ratio of patients who achieved the target corrected Ca level of JSDT at week 30 were similar, 72%, 63%, and 62% in the final low-, medium-, and high-dose subgroups, respectively (Fig 3B, S2 File). In all three subgroups, the ratio of patients who achieved the target P range of JSDT was stable during the evaluation period (64%, 73%, and 63% in the

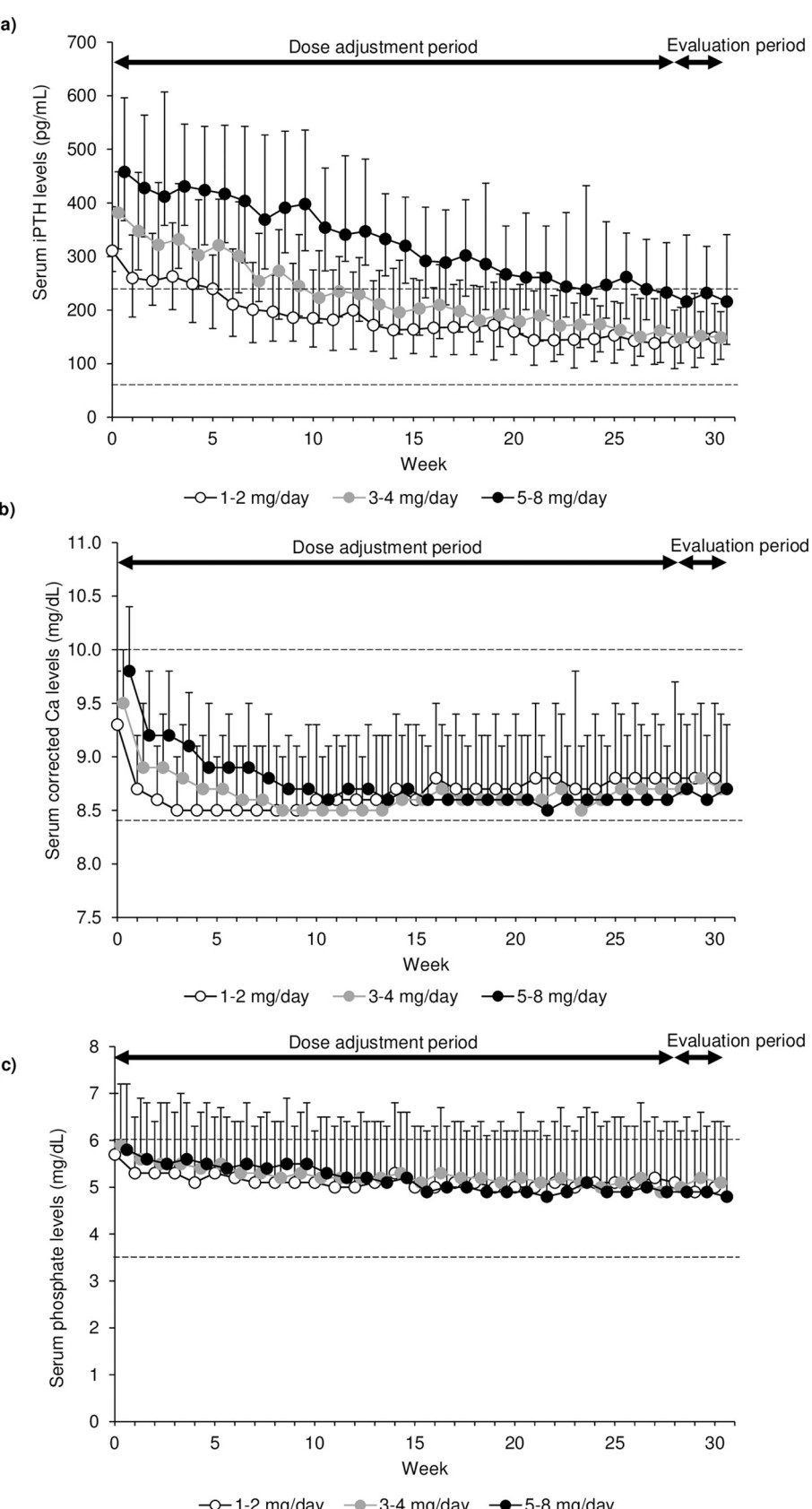

**Fig 2.** Trends in the serum median (IQR) iPTH (a) and mean (SD) corrected Ca (b) and P (c) levels of patients treated with evocalcet. Broken lines indicate the target range of the Japanese Society for Dialysis Therapy. The dose adjustment period was from week 0 to week 28, and the evaluation period was from week 28 to week 30. The efficacy was evaluated during the evaluation period when the final evocalcet doses were administered. Ca, calcium; iPTH, intact parathyroid hormone; IQR, interquartile range; P, phosphate; SD, standard deviation.

low-, medium-, and high-dose subgroups, respectively, at week 30) (Fig 3C, S2 File). The ratio of patients who achieved the target iPTH level of JSDT immediately increased in the final low-dose subgroup after the evocalcet treatment started. As the evocalcet daily dose range increased, the ratio of patients who achieved the target iPTH level of JSDT increased at a slower rate. The ratio of patients who achieved the target corrected Ca level of JSDT decreased similarly from week 0 in the three subgroups because the corrected Ca levels had decreased below the target range of JSDT in some patients. In all three subgroups, the ratio of patients who achieved the target P range of JSDT was stable throughout the study period.

In the low-dose subgroup, the mean (SD) daily dose of evocalcet was increased at week 3 and remained at a similar level until week 28 (1.32 [0.71] mg), whereas in the medium-dose subgroup (3.32 [0.85] mg), the mean daily dose was slightly increased until week 20, and in the high-dose subgroup (6.31 [1.54] mg), the mean daily dose gradually increased until the end of the dose adjustment period (S1 Fig, S3 File).

At week 28, 259 patients were included in the full analysis set (101, 75, and 83 patients in the final low-, medium-, and high-dose subgroups, respectively) (Table 2). When stratified by final evocalcet dose range, patients in the high-dose subgroup were significantly younger and had a longer dialysis duration and less diabetic nephropathy than those in other subgroups. A

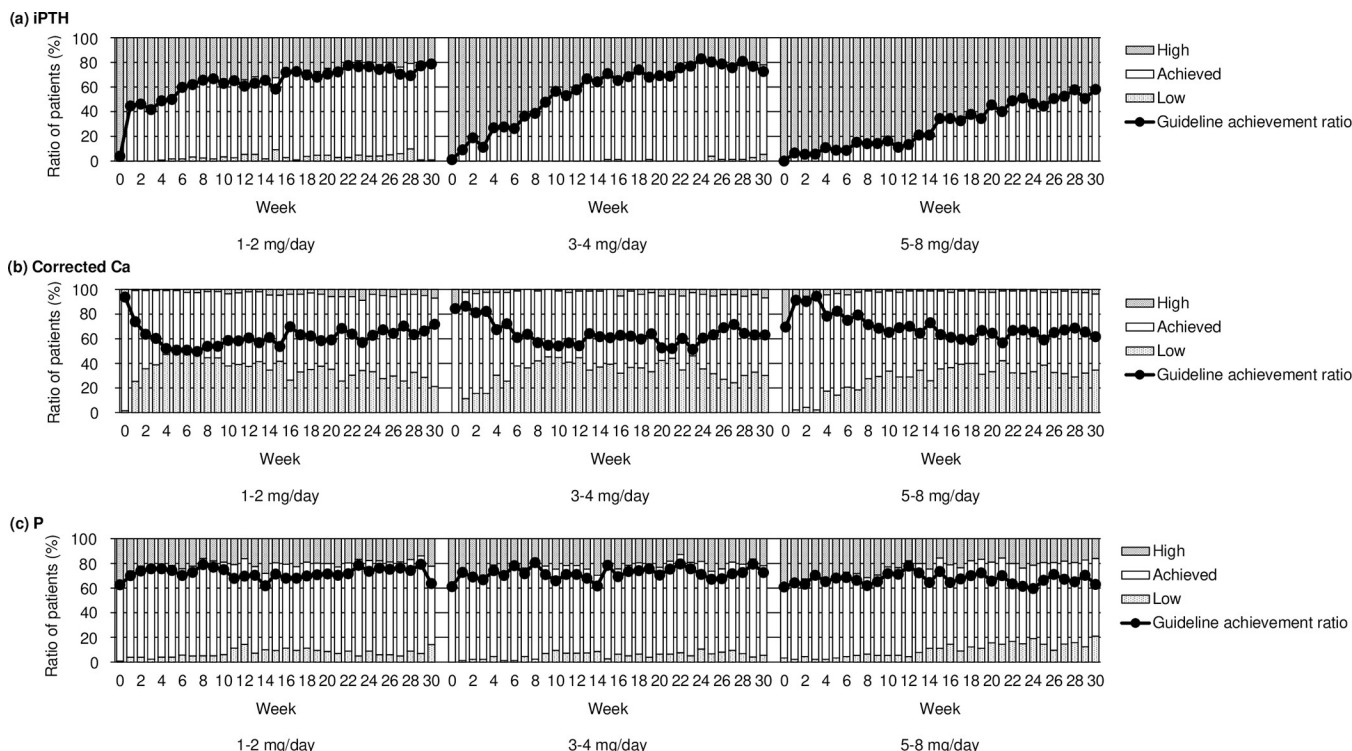

**Fig 3.** The ratio of patients with serum iPTH (a), corrected Ca (b), and P (c) levels above, within, and below the Japanese Society for Dialysis Therapy target ranges. The dose adjustment period was from week 0 to week 28, and the evaluation period was from week 28 to week 30. The efficacy was evaluated during the evaluation period when the final evocalcet doses were administered. Ca, calcium; iPTH, intact parathyroid hormone; P, phosphate.

**Table 2. Patient characteristics at week 28 in each subgroup stratified by the final evocalcet dosages.**

| Parameter | | ALL | Final evocalcet dosages (mg/day) | | | p-value |
|---|---|---|---|---|---|---|
| | | | 1–2 | 3–4 | 5–8 | |
| | | n = 259 | n = 101 | n = 75 | n = 83 | |
| Sex, n (%) | Male | 182 (70) | 74 (73) | 52 (69) | 56 (68) | 0.68 |
| Age, years | | 61 ± 11 | 65 ± 11 | 59 ± 11 | 58 ± 11 | <0.001 |
| Height, cm | | 163 ± 8 | 161 ± 8 | 165 ± 8 | 164 ± 9 | 0.006 |
| Dry weight, kg | | 62 ± 14 | 61 ± 13 | 64 ± 16 | 63 ± 14 | 0.30 |
| Primary kidney disease, n (%) | DN | 67 (26) | 34 (34) | 18 (24) | 15 (18) | <0.001 |
| | CGN | 107 (41) | 29 (29) | 30 (40) | 48 (58) | |
| | NS | 31 (12) | 14 (14) | 5 (7) | 12 (15) | |
| | Other | 54 (21) | 24 (24) | 22 (29) | 8 (10) | |
| Duration of dialysis, months | | 128 ± 88 | 97 ± 74 | 135 ± 95 | 158 ± 86 | <0.001 |
| Type of dialysis, n (%) | HD | 205 (79) | 87 (86) | 61 (81) | 57 (69) | 0.012 |
| | HDF | 54 (21) | 14 (14) | 14 (19) | 26 (31) | |
| Dialysis efficiency (spKt/V) | | 1.49 ± 0.30 | 1.45 ± 0.28 | 1.51 ± 0.29 | 1.51 ± 0.32 | 0.27 |
| Dialysate Ca concentration, mEq/L | | 2.77 ± 0.20 | 2.77 ± 0.19 | 2.76 ± 0.22 | 2.79 ± 0.21 | 0.70 |
| Prior cinacalcet use, n (%) | Yes | 158 (61) | 43 (43) | 46 (61) | 69 (83) | <0.001 |
| Intravenous VDRA use, n (%) | Yes | 139 (54) | 46 (46) | 43 (57) | 50 (60) | 0.10 |
| P binder use, n (%) | Yes | 240 (93) | 90 (89) | 70 (93) | 80 (96) | 0.16 |
| Serum iPTH level, pg/mL | | 157 (108, 253) | 141 (91, 200) | 148 (101, 205) | 216 (150, 340) | <0.001 |
| Serum corrected Ca level, mg/dL | | 8.7 ± 0.7 | 8.8 ± 0.9 | 8.7 ± 0.7 | 8.7 ± 0.6 | 0.68 |
| Serum P level, mg/dL | | 5.0 ± 1.4 | 5.1 ± 1.3 | 5.0 ± 1.3 | 4.9 ± 1.6 | 0.77 |
| Serum BAP level, μg/L | | 17 ± 12 | 15 ± 12 | 17 ± 12 | 20 ± 11 | 0.039 |
| Serum TRACP-5b level, mU/dL | | 497 ± 316 | 430 ± 284 | 456 ± 280 | 615 ± 352 | <0.001 |
| Serum P1NP level, μg/L | | 243 (158, 393) | 193 (133, 320) | 230 (147, 367) | 344 (224, 505) | <0.001 |
| Serum iFGF23 level, pg/mL | | 6,570 (1,500, 16,300) | 6,570 (2,060, 14,200) | 7,920 (1,140, 17,500) | 6,420 (1,800, 18,400) | 0.96 |
| Maximum PTG volume, mm$^3$ | | 322 ± 616 | 231 ± 562 | 249 ± 314 | 479 ± 818 | 0.057 |

Data are expressed as the mean (standard deviation), median (interquartile range), or number (percentage). The statistical significance of differences for categorical variables was estimated using the $\chi^2$ test and continuous variables were assessed using the Kruskal–Wallis test and analysis of variance.

BAP, bone-specific alkaline phosphatase; Ca, calcium; CGN, chronic glomerulonephritis; DN, diabetic nephropathy; HD, hemodialysis; HDF, hemodiafiltration; iFGF23, intact fibroblast growth factor 23; iPTH, intact parathyroid hormone; NS, nephrosclerosis; P, phosphate; P1NP, procollagen type 1 *N*-terminal propeptide; PTG, parathyroid gland; TRACP-5b, tartrate-resistant acid phosphatase-5b; VDRA, vitamin D receptor activator.

higher percentage of patients were on hemodiafiltration in the high-dose subgroup than in the other subgroups. A significantly higher percentage of patients in the high-dose subgroup were treated with cinacalcet before screening. The serum levels of iPTH, BAP, TRACP-5b, and P1NP were significantly higher in the high-dose subgroup than in the other subgroups (p < 0.001 in all cases) at week 28. In contrast, the serum levels of corrected Ca, P, and iFGF23, and maximum PTG volume were not significantly different in the three subgroups.

## Safety

From week 28 to week 30, the incidence of hypocalcemia-related ADRs was 1.2% (3/259 patients) in all patients, and the incidence of gastrointestinal tract-related ADRs in all patients was 0.4% (1/259 patients) (S2 Table). No dose dependence was found in the incidences of hypocalcemia-related and gastrointestinal tract-related ADRs. From week 0 to week 28, ADRs developed in 45% (142/317 patients) of all subjects and in 46% (62/135), 47% (42/90), and 41% (38/92) of subjects in the final low-, medium-, and high-dose subgroups, respectively. The

incidence of hypocalcemia-related ADRs was 19% (59/317 patients) in all patients, and the highest incidence rate was found in the low-dose subgroup. The incidence of gastrointestinal tract-related ADRs in all patients was 13% (41/317 patients), no dose dependence was found.

## Analysis for pre-treatment predictive factors associated with final evocalcet dosages

In univariate analysis, pre-treatment patient characteristics that showed significant associations with increased final evocalcet dose ranges included age, primary kidney disease (diabetic nephropathy), duration of dialysis, type of dialysis, prior cinacalcet use, intravenous VDRA use and P binders (number of types), maximum PTG volume, and serum levels of iPTH, corrected Ca, P1NP, and iFGF23 (S3 Table). Baseline characteristics that showed significant associations, except for baseline iPTH levels that had an extremely high upper limit (>1000.00), were further subjected to multivariate analysis.

Of the pre-treatment patient characteristics that showed significant associations with medium- and/or high-dose ranges of final evocalcet dosages by univariate analysis, multivariate analysis further demonstrated that younger age, higher serum corrected Ca level, and larger maximum PTG volume were significantly associated with the medium-dose range ($p < 0.05$ in all cases, Table 3). In addition to the factors identified in the final medium-dose subgroup,

**Table 3. Multivariate analyses for factors associated with medium and/or high final evocalcet dosages using patient pre-treatment characteristics as variables.**

| Variables | Final evocalcet dosages | Multivariate analysis | | |
|---|---|---|---|---|
| | | OR | 95% CI | *p*-value |
| Age (year) | Medium | 0.96 | 0.93–0.99 | 0.004 |
| | High | 0.95 | 0.92–0.99 | 0.007 |
| Primary kidney disease (DN) | Medium | 0.63 | 0.32–1.24 | 0.18 |
| | High | 0.54 | 0.24–1.25 | 0.15 |
| Duration of dialysis (year) | Medium | 1.03 | 0.98–1.08 | 0.22 |
| | High | 1.03 | 0.98–1.09 | 0.23 |
| Type of dialysis (HD) | Medium | 0.86 | 0.38–1.93 | 0.71 |
| | High | 0.59 | 0.24–1.43 | 0.24 |
| Prior cinacalcet use | Medium | 1.38 | 0.73–2.63 | 0.32 |
| | High | 2.83 | 1.25–6.40 | 0.01 |
| Intravenous VDRA use | Medium | 0.65 | 0.35–1.19 | 0.16 |
| | High | 0.57 | 0.28–1.18 | 0.13 |
| P binder use (number of types) | Medium | 1.15 | 0.79–1.68 | 0.46 |
| | High | 1.40 | 0.91–2.16 | 0.13 |
| Serum corrected Ca level (mg/dL) | Medium | 2.18 | 1.06–4.47 | 0.03 |
| | High | 4.08 | 1.83–9.13 | <0.001 |
| Serum P1NP level (log) | Medium | 0.52 | 0.52–5.32 | 0.39 |
| | High | 13.87 | 3.38–56.95 | <0.001 |
| Serum iFGF23 level (log) | Medium | 1.19 | 0.53–2.68 | 0.67 |
| | High | 3.30 | 1.30–8.83 | 0.02 |
| Maximum PTG volume (≥200 mm$^3$) | Medium | 2.39 | 1.06–5.40 | 0.04 |
| | High | 5.27 | 2.10–13.21 | <0.001 |

A logistic regression model was applied to assess the week 0 characteristics of patients who required the medium-dose or high-dose range of final evocalcet dosages. All of the variables listed in Tables 1 and 2 were entered into the model.

Evocalcet dose range: Medium, 3–4 mg/day; and high, 5–8 mg/day. Ca, calcium; CI, confidence interval; DN, diabetic nephropathy; HD, hemodialysis; iFGF23, intact fibroblast growth factor 23; OR, odds ratio; P, phosphate; P1NP, procollagen type 1 N-terminal propeptide; PTG, parathyroid gland; VDRA, vitamin D receptor activator.

prior cinacalcet use and higher serum P1NP and iFGF23 levels were significantly associated with the final high-dose range ($p < 0.05$ in all cases).

## Discussion

Our results demonstrated that serum iPTH levels in the final high-dose subgroup were improved to a lesser degree than those of the low- and medium-dose subgroups, despite the serum corrected Ca and P levels decreasing to similar levels in all three subgroups. The present post hoc analysis also identified larger maximum PTG, higher serum iPTH and corrected Ca levels, and younger age as predictive factors for patients who required a final medium-dose range of evocalcet. In addition to the predictive factors for patients requiring the medium dose, prior cinacalcet use and higher serum levels of P1NP and iFGF23 were identified as predictive factors for patients who required a final high-dose range. These results suggested that final higher doses of evocalcet were required in patients with severe SHPT. This finding is supported by a previous study demonstrating that enlarged PTGs related to SHPT had a decreased expression of Ca-sensing receptors and reduced sensitivity to Ca in PTH regulation [20]. The present results further indicated that SHPT was more advanced in patients who required the high-dose range of evocalcet, because bone turnover was increased. Although any potential impact of differences regarding prior treatment with cinacalcet and VDRA should be considered, SHPT appeared to be controlled at a low evocalcet dose if the treatment was initiated before the set point of the PTH-Ca curve shifted largely to the upper right, which was caused by the progression of SHPT. Indeed, many patients with median iPTH ≥450 pg/mL and mean corrected Ca ≥9.8 mg/dL at week 0 were in the final high-dose subgroup, and those with ≤300 pg/mL and ≤9.3 mg/dL, respectively, were in the low-dose subgroup.

Elevated PTH levels lead to increased bone turnover, which induces further bone loss [21,22] and Ca and P release from bones, followed by vascular calcification. Moreover, cardiac hypertrophy is caused by vascular calcification and excess FGF23 secretion related to uncontrolled SHPT [23–25] and excess PTH induces cachexia [26]. Because these factors affect morbidity and mortality, it is critical to initiate evocalcet treatment in the early stages of SHPT to control the serum iPTH, corrected Ca, and P levels as adequately as possible for a better prognosis.

A previous study reported that the number of tablets taken by patients significantly affected their adherence to that medication [27]. This finding may also apply to evocalcet in the clinical setting because patients are required to take more tablets as the dose of evocalcet increases. Thus, it is critical to initiate evocalcet treatment in the early stages of SHPT for better adherence, although there is no dose dependence for ADRs.

Of note, patients included in this study were treated for up to 30 weeks [4]. A previous study of evocalcet showed that 52 weeks of treatment continued to improve iPTH levels [17]. Another retrospective study showed that cinacalcet for up to 72 weeks decreased iPTH levels to similar levels among patients with baseline iPTH levels of 301–500, 501–800, and >800 pg/mL [28]. These results suggest that iPTH levels might decrease further if patients are treated for a longer period, even if they are in the high-dose subgroup. However, it is not recommended to expose patients to excess PTH for a long duration for the reasons described above. Therefore, it is critical to initiate evocalcet treatment in the early stages of SHPT to control serum iPTH and corrected Ca levels as adequately as possible.

This study had some limitations. First, increasing the evocalcet dose might be difficult because serum-corrected Ca levels remained low because it was prohibited to increase the VDRA dose. Second, an evocalcet dose increase over 8 mg/day was prohibited, although the dose could be increased to a maximum 12 mg/day in a clinical setting. Third, an increase in

evocalcet dose is allowed 2 weeks after a previous dose increase in real clinical settings, whereas it was 3 weeks in this study [12]. Therefore, evocalcet might control serum iPTH and corrected Ca levels better for a shorter period in real clinical settings where the dosages of VDRA can be adjusted in combination with evocalcet. Fourth, this study had a relatively small sample size.

In conclusion, patients with younger age, high serum levels of iPTH, corrected Ca, iFGF23, and bone turnover markers, large maximum PTG volume, and prior cinacalcet use (indicating more advanced SHPT) were significantly more likely to require a final high-dose range of evocalcet. Additionally, the serum iPTH level in the final high-dose subgroup was improved to a lesser degree than the levels in the low- and medium-dose subgroups. The present study revealed that it is critical to start evocalcet treatment in the early stages of SHPT to better control chronic kidney disease-related mineral and bone disorders. Further studies in real clinical settings using a larger sample size are needed to verify our findings.

## Supporting information

**S1 Fig. Time course of changes in the mean (standard deviation) evocalcet dose.**
(TIF)

**S1 Table. List of Institutional Review Boards.**
(PDF)

**S2 Table. Incidence of major adverse drug reactions.**
(PDF)

**S3 Table. Univariate analyses of factors associated with medium and/or high final evocalcet dosages using patient pre-treatment characteristics as variables.**
(PDF)

**S1 File. Serum iPTH, corrected Ca and P levels in patients with SHPT treated with evocalcet for 30 weeks.**
(PDF)

**S2 File. Ratio of patients with serum iPTH, corrected Ca and P levels above, within, and below the target ranges of Japanese Society for Dialysis Therapy for 30 weeks.**
(PDF)

**S3 File. Changes in the mean (SD) evocalcet dose for 30 weeks.**
(PDF)

**S4 File. List of collaborators.**
(PDF)

## Acknowledgments

The authors thank all the collaborators for their participation in the study (see S4 File), Ms. Shiori Yasui, Mr. Taichi Mizogui, and Dr. Shinjo Yada of A2 Healthcare Corporation for statistical analyses, and ASCA Corporation for assisting with the writing and editing of the manuscript.

## Ethical approval

The study was approved by the ethics committee at each study site (S1 Table).

## Author Contributions

**Conceptualization:** Masanori Tokumoto, Shin Tokunaga, Shinji Asada, Yuichi Endo, Masafumi Fukagawa, Tadao Akizawa.

**Data curation:** Yuichi Endo.

**Investigation:** Shin Tokunaga, Shinji Asada, Yuichi Endo.

**Methodology:** Masanori Tokumoto, Shin Tokunaga, Shinji Asada, Yuichi Endo, Noriaki Kurita.

**Project administration:** Shin Tokunaga.

**Supervision:** Masafumi Fukagawa, Tadao Akizawa.

**Visualization:** Masanori Tokumoto, Shin Tokunaga.

**Writing – original draft:** Masanori Tokumoto, Shin Tokunaga.

**Writing – review & editing:** Shinji Asada, Yuichi Endo, Noriaki Kurita, Masafumi Fukagawa, Tadao Akizawa.

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
