## [Decision Letter · Decision Letter 0]

23 Feb 2022

PONE-D-21-08923Predictive factors requiring high-dose evocalcet in hemodialysis patients with secondary hyperparathyroidismPLOS ONE

Dear Dr. Tokumoto,

Thank you for submitting your manuscript to PLOS ONE. After careful consideration, we feel that it has merit but does not fully meet PLOS ONE’s publication criteria as it currently stands. Therefore, we invite you to submit a revised version of the manuscript that addresses the points raised during the review process.

Please note that both referees raised several issues (see their reports below) that should be addressed thoroughly. Besides, reviewer #2 was somewhat skeptic about the robustness of the statistical analysis and the rationale of some analyses. I'd suggest addressing these comments specifically. 

We look forward to receiving your revised manuscript.

Kind regards,

Gianpaolo Reboldi, MD, MSc, PhD

Academic Editor

PLOS ONE

Journal Requirements:

2. Thank you for stating the following in the Competing Interests section: "MT received consulting fees from KKC and Ono Pharmaceutical, and lecture fees from KKC, Chugai Pharmaceutical, Bayer, Kissei Pharmaceutical, Torii Pharmaceutical, Fuso Pharmaceutical, and Ono Pharmaceutical. ST, SA, and YE are employees of KKC. NK received lecture fees and travel grants for research meetings from KKC, which are not connected to this research. MF received consulting fees from KKC and Ono Pharmaceutical, lecture fees from KKC, Bayer, Torii Pharmaceutical and Ono Pharmaceutical, and grants from KKC and Bayer. TA received consulting fees from KKC, Astellas Pharma, Bayer, Fuso Pharmaceutical, Japan Tobacco, Ono Pharmaceutical, Sanwa Chemical, Otsuka, GSK, and NIPRO, and lecture fees from KKC, Chugai Pharmaceutical, Bayer, Kissei Pharmaceutical, Torii Pharmaceutical, and Ono Pharmaceutical."

We note that you received funding from a commercial source: Kyowa Kirin Co., Ltd, Ono Pharmaceutical, Chugai Pharmaceutical, Bayer, Kissei Pharmaceutical, Torii Pharmaceutical, and Fuso Pharmaceutical

5. Please remove all personal information, ensure that the data shared are in accordance with participant consent, and re-upload a fully anonymized data set. 

Reviewers' comments:

Reviewer's Responses to Questions

**Comments to the Author**

1. Is the manuscript technically sound, and do the data support the conclusions?

Reviewer #1: Yes

Reviewer #2: No

2. Has the statistical analysis been performed appropriately and rigorously? 

Reviewer #1: Yes

Reviewer #2: No

3. Have the authors made all data underlying the findings in their manuscript fully available?

Reviewer #1: Yes

Reviewer #2: No

4. Is the manuscript presented in an intelligible fashion and written in standard English?

Reviewer #1: Yes

Reviewer #2: No

5. Review Comments to the Author

Reviewer #1: General comments:

This is a post-hoc analysis of a phase 3 randomized clinical trial which compared head-to-head the efficacy and safety of oral cinacalcet hydrochloride to evocalcet. The post-hoc analysis included 313 hemodialysis patients and looked at the baseline clinical and biochemical parameters determining the response and the use of higher doses of the oral calcimimetic evocalcet. The authors found that at the end of study, median serum PTH level was higher and achievement ratio of the Japanese Society for Dialysis Therapy target range was lower in the high-dose subgroup (216 pg/mL, 58%, respectively) than other subgroups (low: 149 pg/mL, 79%; medium: 149 pg/mL, 73%, respectively). Younger age, prior cinacalcet use, higher levels of serum PTH, corrected calcium, procollagen type 1 N-terminal propeptide, intact fibroblast growth factor-23, and larger maximum parathyroid gland volume were significantly associated with high dose evocalcet (p < 0.05 in all cases).

The authors should consider the following comments and questions.

Specific comments:

1- Was the calcium concentration in the dialysate kept stable during the study? It would be of interest knowing which was the mean (median) calcium concentration in the dialysate fluid for each one of the patient groups.

2- How was the compliance to cinacalcet and evocalcet evaluated?

3- Circulating levels of 25OHD3 have been demonstrated to be associated with the response to oral cinacalcet. Did the authors assess circulating levels of 25OD3 in this trial?

Reviewer #2: Here is a list of specific comments. Note: line and page numbering in reviews and comments is based on ruler applied in Editorial Manager-generated PDF.

1. Page 4, lines 59–60: This phrase inferred causality. Because the post-hoc analysis lost the benefit of randomization, I suggest using ‘associate’ instead of using “predict”.

2. Page 6, lines 99–101: I suggest indicating the categorized evocalcet dose at week 28 was the primary exposure of interest.

3. Page 7, line 110: Because the primary exposure of interest was determined at week 28, I suggest clarifying in the manuscript that the baseline for endpoints was week 28.

4. Page 7, lines 110–111: I suggest not mentioning “in each subgroup stratified by evocalcet dose range” in the Endpoints section.

5. Page 7, lines 111–112: This sentence was also irrelevant in the Endpoints section.

6. Page 7, lines 122–126: I suggest clarifying which characteristics were collected at week 28. If a characteristic was collected at week 28, I suggest referring the week-28 value as the “baseline” value.

7. Page 7, line 127: Did ratios refer to percentages? If so, I suggest replacing ratios with percentages.

8. Page 7, lines 127–128: I suggest removing “and summary statistics were calculated for continuous data” because it was redundant given the following sentence.

9. Page 7, line 129: Please write ‘were calculated for continuous data’.

10. Page 8, line 131: Following Comment #3 above, I suggest limiting to the time points after week 28.

11. Page 8, line 133: Following Comment #3 above, I suggest starting from week 28.

12. Page 8, lines 136–140: I did not understand the rationale of this analysis. The evocalcet categories were previously defined and described as the primary exposure of interest. However, the multinomial logistic used the evocalcet categories as the outcome without any justification.

13. Page 8, lines 141–143: Similarly, the safety endpoints and follow-up time should start from week 28 to week 30.

14. Page 8, lines 144–146: I suggest relocating this sentence to the end of the second paragraph of this section.

15. Page 8, lines 150–151: I suggest including a figure depicting the patient flow from the randomization to the safety and full analysis sets.

6. PLOS authors have the option to publish the peer review history of their article (what does this mean?). If published, this will include your full peer review and any attached files.

Reviewer #1: No

Reviewer #2: No

---

## [Author Response · Author response to Decision Letter 0]

13 Apr 2022

Responses to review comments

Journal Requirements:

Response: We have confirmed that our manuscript is formatted according to the PLOS ONE style requirements.

2. Thank you for stating the following in the Competing Interests section: "MT received consulting fees from KKC and Ono Pharmaceutical, and lecture fees from KKC, Chugai Pharmaceutical, Bayer, Kissei Pharmaceutical, Torii Pharmaceutical, Fuso Pharmaceutical, and Ono Pharmaceutical. ST, SA, and YE are employees of KKC. NK received lecture fees and travel grants for research meetings from KKC, which are not connected to this research. MF received consulting fees from KKC and Ono Pharmaceutical, lecture fees from KKC, Bayer, Torii Pharmaceutical and Ono Pharmaceutical, and grants from KKC and Bayer. TA received consulting fees from KKC, Astellas Pharma, Bayer, Fuso Pharmaceutical, Japan Tobacco, Ono Pharmaceutical, Sanwa Chemical, Otsuka, GSK, and NIPRO, and lecture fees from KKC, Chugai Pharmaceutical, Bayer, Kissei Pharmaceutical, Torii Pharmaceutical, and Ono Pharmaceutical."

We note that you received funding from a commercial source: Kyowa Kirin Co., Ltd, Ono Pharmaceutical, Chugai Pharmaceutical, Bayer, Kissei Pharmaceutical, Torii Pharmaceutical, and Fuso Pharmaceutical

Response: The sentence “This does not alter our adherence to PLOS ONE policies on sharing data and materials.” has been added after the COI statement in the second paragraph in the cover letter. 

Response: In accordance with your suggestion, the following three files have been added as Supporting information to provide the underlying data used to obtain the percentage of patients achieving the guideline target range for serum iPTH, corrected Ca, and P levels, and changes in the evocalcet dose in our manuscript.

S1 File. Changes in the mean (SD) evocalcet dose for 30 weeks.

S2 File. Serum iPTH, corrected Ca and P levels in patients with SHPT treated with evocalcet for 30 weeks.

S3 File. The percentage of patients with serum iPTH, corrected Ca and P levels above, within, and below the Japanese Society for Dialysis Therapy target ranges for 30 weeks.

Response: We confirm that the full names of the ethics committees have been presented in “S1 Table. List of Institutional Review Boards”, and a full ethics statement including the informed written consent has been included in the Study design section (Page 6, Line 111- Page 7, Line 115). 

5. Please remove all personal information, ensure that the data shared are in accordance with participant consent, and re-upload a fully anonymized data set. 

Response: We confirm that no personal information has been included in any part of our manuscript, and that any participant information has been handled as anonymized data according to the consent form. 

Reviewers' comments:

Reviewer's Responses to Questions

Reviewer #1: General comments:

This is a post-hoc analysis of a phase 3 randomized clinical trial which compared head-to-head the efficacy and safety of oral cinacalcet hydrochloride to evocalcet. The post-hoc analysis included 313 hemodialysis patients and looked at the baseline clinical and biochemical parameters determining the response and the use of higher doses of the oral calcimimetic evocalcet. The authors found that at the end of study, median serum PTH level was higher and achievement ratio of the Japanese Society for Dialysis Therapy target range was lower in the high-dose subgroup (216 pg/mL, 58%, respectively) than other subgroups (low: 149 pg/mL, 79%; medium: 149 pg/mL, 73%, respectively). Younger age, prior cinacalcet use, higher levels of serum PTH, corrected calcium, procollagen type 1 N-terminal propeptide, intact fibroblast growth factor-23, and larger maximum parathyroid gland volume were significantly associated with high dose evocalcet (p < 0.05 in all cases).

The authors should consider the following comments and questions.

Specific comments:

1- Was the calcium concentration in the dialysate kept stable during the study? It would be of interest knowing which was the mean (median) calcium concentration in the dialysate fluid for each one of the patient groups.

Response: It was prohibited to change the prescribed dialysis conditions, including the calcium concentration in the dialysate until week 30 (or at the time of discontinuation). The mean ± standard deviation (SD) dialysate calcium concentration at baseline (week 0) was 2.78 ± 0.19 mEq/L in the low-dose (1–2 mg), 2.77 ± 0.21 mEq/L in the medium-dose (3–4 mg), and 2.79 ± 0.20 mEq/L in the high-dose range (5–8 mg).

Details of the method used to control the dialysate conditions and data of dialysate calcium concentrations have been added to the “Previous phase 3 head-to-head comparison study” section of the Materials and methods (Page 5, Lines 85-88), Results (Page 9, Lines 169-171), and Table 1 (Page 10). 

2- How was the compliance to cinacalcet and evocalcet evaluated?

Response: The medication adherence rate was calculated using the following formula:

Adherence rate (%) = 100 × number of days administered as prescribed/total number of prescribed days.

We found a high compliance to the medications according to the adherence rate for evocalcet (mean ± SD: 98.51 ± 6.24%) and cinacalcet (98.91 ± 2.24%). The results have been added to the “Previous phase 3 head-to-head comparison study” section of the Materials and methods (Page 6, Lines 99-100). 

3- Circulating levels of 25OHD3 have been demonstrated to be associated with the response to oral cinacalcet. Did the authors assess circulating levels of 25OD3 in this trial?

Response: Thank you for providing the information regarding the association between 25-hydroxyvitamin D levels and the response to cinacalcet (Zheng CM, et al. Nutrients. 2018;10:196). The circulating levels of 25-hydroxyvitamin D were not measured in the previous phase 3 head-to-head cinacalcet and evocalcet study (Fukagawa, et al., Kidney Int. 2018;94:818–825). Therefore, we do not have any data to examine an association between 25-hydroxyvitamin D levels and the response to evocalcet. 

Reviewer #2: Here is a list of specific comments. Note: line and page numbering in reviews and comments is based on ruler applied in Editorial Manager-generated PDF.

1. Page 4, lines 59–60: This phrase inferred causality. Because the post-hoc analysis lost the benefit of randomization, I suggest using ‘associate’ instead of using “predict”.

Response: This has been revised to “which factors are associated with patients who…” on Page 4, Line 62. 

2. Page 6, lines 99–101: I suggest indicating the categorized evocalcet dose at week 28 was the primary exposure of interest.

Response: In this study, we aimed to identify factors to predict which patients will require a high-dose range of evocalcet for SHPT treatment. To achieve this, we analyzed efficacy and safety, and patients’ baseline characteristics associated with the final dose of evocalcet according to the finally-administered evocalcet dose range using the data set obtained in the previous phase 3 head-to-head, active-controlled, randomized, double-blind, comparison study. As you will find in the study design of the previous phase 3 study below, the last dose adjustment was carried out at week 28.

　 

(Fukagawa, et al., Kidney Int. 2018;94:818–825)

Therefore, this study was designed (1) to evaluate the efficacy and safety of evocalcet from week 0 to week 30 according to the evocalcet dose range administered to patients at week 28, and (2) to explore the baseline characteristics (week 0) of patients who required the high-dose evocalcet range in week 28.

(1) The efficacy (serum iPTH, cCa, and P, as well as the percentage of patients achieving the JSDT guidelines target ranges) and safety from week 0, when the treatment with evocalcet was initiated, until the end of the study, were evaluated according to the evocalcet dose administered to patients at week 28. 

(2) Baseline characteristics of patients who required a final high-dose range of evocalcet during the dose adjustment period, despite a starting low dose of 1 or 2 mg evocalcet, were identified, so that a group of patients who might require a high-dose range of evocalcet could be predicted. 

Therefore, the evocalcet dose administered to patients at week 28 was used to stratify the data, and the baseline of each subgroup was set at week 0 when the administration of evocalcet was initiated. 

3. Page 7, line 110: Because the primary exposure of interest was determined at week 28, I suggest clarifying in the manuscript that the baseline for endpoints was week 28.

Response: As you will find in our response to comment #2, the baseline for the evaluation of endpoints was set at week 0 (start of treatment), not week 28.

4. Page 7, lines 110–111: I suggest not mentioning “in each subgroup stratified by evocalcet dose range” in the Endpoints section.

Response: The stratification of the data according to the three ranges of daily evocalcet dose at week 28 was described on Page 6, Lines 108-110 in the Study design section. Therefore, “in each subgroup stratified by evocalcet dose range” has been deleted, accordingly. 

5. Page 7, lines 111–112: This sentence was also irrelevant in the Endpoints section.

Response: Indeed, it is irrelevant to describe the analysis in the Endpoints section. Therefore, the sentence “Risk factors associated with an evocalcet dose increase and trends in the evocalcet dose were also evaluated for each dose range.” has been deleted. 

6. Page 7, lines 122–126: I suggest clarifying which characteristics were collected at week 28. If a characteristic was collected at week 28, I suggest referring the week-28 value as the “baseline” value.

Response: As explained in our response to comment #2, the baseline data used for the analyses in this study were collected at week 0 when the evocalcet treatment was initiated.

7. Page 7, line 127: Did ratios refer to percentages? If so, I suggest replacing ratios with percentages.

Response: As pointed out, the achievement ratio is shown as a percentage in the JSDT guidelines. Therefore, “ratio” has been replaced with “percentage” in the main text and figures. 

8. Page 7, lines 127–128: I suggest removing “and summary statistics were calculated for continuous data” because it was redundant given the following sentence.

Response: In accordance with your suggestion, “and summary statistics were calculated for continuous data” has been deleted. 

9. Page 7, line 129: Please write ‘were calculated for continuous data’.

Response: This has been revised to “were calculated for continuous data” on Page 8, Line 139. 

10. Page 8, line 131: Following Comment #3 above, I suggest limiting to the time points after week 28.

Response: As you will find in our response to comment #2, the baseline was set at week 0 (start of the treatment), not week 28, for all subgroups. Therefore, trends in the iPTH, cCa, and P levels were analyzed from week 0 to week 30. 

11. Page 8, line 133: Following Comment #3 above, I suggest starting from week 28.

Response: In response to comment 2, we explained that the baseline of subgroups was not week 28 but week 0 when the treatment with evocalcet was initiated. Therefore, trends in the percentages of patients who achieved JSDT target ranges of iPTH, cCa, and P levels were analyzed from week 0 to week 30. 

12. Page 8, lines 136–140: I did not understand the rationale of this analysis. The evocalcet categories were previously defined and described as the primary exposure of interest. However, the multinomial logistic used the evocalcet categories as the outcome without any justification.

Response: As we responded to comment 2, the study was designed to evaluate the efficacy and safety of evocalcet from week 0 to week 30 according to the evocalcet dose range that patients received at week 28, and to identify the baseline characteristics (week 0) of patients who required the high-dose evocalcet range at week 28.

13. Page 8, lines 141–143: Similarly, the safety endpoints and follow-up time should start from week 28 to week 30.

Response: As we mentioned in our response to comment #12, the study was designed to evaluate the evocalcet dose-dependent (as of week 28) efficacy and safety during week 0 to week 30, and identify patients’ baseline characteristics associated with high-dose evocalcet required at week 28.

14. Page 8, lines 144–146: I suggest relocating this sentence to the end of the second paragraph of this section.

Response: As pointed out, the sentence “Statistical differences for categorical variables were estimated using the χ2 test (two-tailed), and continuous variables were assessed using the Kruskal–Wallis test and analysis of variance (two-tailed) according to the distribution.” has been moved to the end of the second paragraph of the “Statistical methods” section. 

15. Page 8, lines 150–151: I suggest including a figure depicting the patient flow from the randomization to the safety and full analysis sets.

Response: The data set used in this study was reported in the previous phase 3 head-to-head, active-controlled, randomized, double-blind, comparison study, which is cited in the Materials and methods and Results sections of our manuscript. The patient flow is shown below for your information (Fukagawa, et al., Kidney Int. 2018;94:818–825). 

6. PLOS authors have the option to publish the peer review history of their article (what does this mean?). If published, this will include your full peer review and any attached files.

---

## [Decision Letter · Decision Letter 1]

2 May 2022

PONE-D-21-08923R1Predictive factors requiring high-dose evocalcet in hemodialysis patients with secondary hyperparathyroidismPLOS ONE

Dear Dr. Tokumoto,

Thank you for submitting your manuscript to PLOS ONE. After careful consideration, we feel that it has merit but does not fully meet PLOS ONE’s publication criteria as it currently stands.  While reviewer #1 was satisfied by the revised version, reviewer #2 remarked issues regarding the the design and analysis plan that must be addressed (see details below).

Therefore, we invite you to submit a revised version of the manuscript that addresses the points raised by reviewer #2

We look forward to receiving your revised manuscript.

Kind regards,

Gianpaolo Reboldi, MD, MSc, PhD

Academic Editor

PLOS ONE

Reviewers' comments:

Reviewer's Responses to Questions

**Comments to the Author**

1. If the authors have adequately addressed your comments raised in a previous round of review and you feel that this manuscript is now acceptable for publication, you may indicate that here to bypass the “Comments to the Author” section, enter your conflict of interest statement in the “Confidential to Editor” section, and submit your "Accept" recommendation.

Reviewer #1: All comments have been addressed

Reviewer #2: (No Response)

2. Is the manuscript technically sound, and do the data support the conclusions?

Reviewer #1: Yes

Reviewer #2: Partly

3. Has the statistical analysis been performed appropriately and rigorously? 

Reviewer #1: Yes

Reviewer #2: No

4. Have the authors made all data underlying the findings in their manuscript fully available?

Reviewer #1: Yes

Reviewer #2: No

5. Is the manuscript presented in an intelligible fashion and written in standard English?

Reviewer #1: Yes

Reviewer #2: Yes

6. Review Comments to the Author

Reviewer #1: They have properly answered to my main questions. The manuscript has been significantly improved.

I do not have further comments.

Reviewer #2: Here is a list of specific comments. Note: line and page numbering in reviews and comments is based on ruler applied in Editorial Manager-generated PDF.

1. Page 6, lines 103–106: I suggest reserving ‘this study’ for the post-hoc analysis and referring the previous phase 3 head-to-head comparison study as ‘the parent study’.

2. Page 6, line 104:

(2a) My apologies for misinterpreting the role of the evocalcet categories in Comment #2 of the previous review.

(2b) There were two roles of the evocalcet categories in this study: (1) as an outcome in identifying factors associated with the evocalcet categories; and (2) as a stratification factor in evaluating efficacy and safety of evocalcet.

(2c) For the second role, stratifying patients based on a post-baseline factor was likely to introduce bias. In this case, the evocalcet categories was not determined until Week 28 whereas the factors that were evaluated were measured at baseline, Week 0. I suggest considering a more careful analytic plan.

(2d) To avoid the aforementioned bias, you may treat Week 28 as baseline. This means that only efficacy and safety endpoints that were measured and collected after Week 28 will be treated as outcomes. Everything collected at or prior to Week 28 could be consider as baseline factor. In this case, the stratified analysis would be OK as long as the outcomes were collected after Week 28.

3. Page 7, line 115: These endpoints were the outcomes for the second role of the evocalcet categories (see Comment #2b above). For the first role, the outcome would be the evocalcet categories. I suggest explicitly defining the outcome for the first role.

4. Page 7, lines 124–127:

(4a) Comment #15 of the previous review was meant to clarify the full analysis set and the safety analysis set in this study. The figure in the response was for the parent study. I suggest including a patient flow diagram for this study even if it would be a copy of the left-hand side of the original figure.

(4b) In addition, the full analysis and safety sets were with regard to the second role of the evocalcet categories (see Comment #2b above).

5. Page 8, lines 134–136: Given Comment #2c above, these comparisons would not be performed unless there was another stratified variable that was defined at baseline. Otherwise, this sentence could be removed.

6. Page 8, lines 143–147: This was only for the first role of the evocalcet categories (see Comment #2b above). The analytic plan for the efficacy endpoints were not included.

7. PLOS authors have the option to publish the peer review history of their article (what does this mean?). If published, this will include your full peer review and any attached files.

Reviewer #1: **Yes: **Pablo URENA TORRES

Reviewer #2: No

---

## [Author Response · Author response to Decision Letter 1]

1 Nov 2022

Reviewer #2: 

1. Page 6, lines 103–106: I suggest reserving ‘this study’ for the post-hoc analysis and referring the previous phase 3 head-to-head comparison study as ‘the parent study’.

Response: Accordingly, “the previous phase 3 head-to-head comparison study” has been replaced with “the parent study.”

2. Page 6, line 104:

(2a) My apologies for misinterpreting the role of the evocalcet categories in Comment #2 of the previous review.

(2b) There were two roles of the evocalcet categories in this study: (1) as an outcome in identifying factors associated with the evocalcet categories; and (2) as a stratification factor in evaluating efficacy and safety of evocalcet.

(2c) For the second role, stratifying patients based on a post-baseline factor was likely to introduce bias. In this case, the evocalcet categories was not determined until Week 28 whereas the factors that were evaluated were measured at baseline, Week 0. I suggest considering a more careful analytic plan.

(2d) To avoid the aforementioned bias, you may treat Week 28 as baseline. This means that only efficacy and safety endpoints that were measured and collected after Week 28 will be treated as outcomes. Everything collected at or prior to Week 28 could be consider as baseline factor. In this case, the stratified analysis would be OK as long as the outcomes were collected after Week 28.

Response: We understand and appreciate the reviewer’s comments. In accordance with these comments, the analytical plan has been changed and additional analyses have been carried out. 

The efficacy and safety analysis plan has been changed to focus on the outcomes in the evaluation period from week 28 to week 30 with baseline at week 28. Accordingly, the efficacy and safety results from week 28 to week 30, as well as patient characteristics in week 28, have been re-analyzed, and the Materials and methods and the Results sections have been revised. The time courses of serum iPTH and corrected Ca levels from week 0 to week 28 are also included as a reference in Figures 2 and 3 to better understand the efficacy of evocalcet during the dose adjustment period.

3. Page 7, line 115: These endpoints were the outcomes for the second role of the evocalcet categories (see Comment #2b above). For the first role, the outcome would be the evocalcet categories. I suggest explicitly defining the outcome for the first role.

Response: In the section “Analysis for pre-treatment predictive factors associated with final evocalcet dosages,” the sentence “Patient characteristics at week 0 associated with medium and/or high final evocalcet dose ranges were retrospectively identified by a logistic regression model” has been added to define the outcome for the first role (Lines 148-149, Page 9). 

4. Page 7, lines 124–127: 

(4a) Comment #15 of the previous review was meant to clarify the full analysis set and the safety analysis set in this study. The figure in the response was for the parent study. I suggest including a patient flow diagram for this study even if it would be a copy of the left-hand side of the original figure.

Response: Accordingly, the patient flow diagram has been added to the manuscript as Fig. 1.

(4b) In addition, the full analysis and safety sets were with regard to the second role of the evocalcet categories (see Comment #2b above).

Response: The identification of pre-treatment predictive factors was analyzed in the full analysis set. Therefore, the following sentence has been added to the first paragraph of the Statistical methods section (Lines 155-156, Page 9).

“Pre-treatment predictive factors associated with final evocalcet dosages were evaluated in the full analysis set.”

5. Page 8, lines 134–136: Given Comment #2c above, these comparisons would not be performed unless there was another stratified variable that was defined at baseline. Otherwise, this sentence could be removed.

Response: To be more specific, these sentences have been changed to “The mean and SD, or median and interquartile range, were calculated for continuous data in all patients as well as in each dose subgroup. The statistical significance of differences in dose subgroups was estimated using the χ2 test (two-tailed) for categorical variables and the Kruskal–Wallis test and analysis of variance (two-tailed) according to the distribution for continuous variables” (Lines 164-169, Page 10).

6. Page 8, lines 143–147: This was only for the first role of the evocalcet categories (see Comment #2b above). The analytic plan for the efficacy endpoints were not included.

Response: The analytical plan for the efficacy endpoints has been updated as follows (Lines 170-175, Pages 10). Note that the data from week 0 to week 30 are presented in each figure, but the data from week 28 to week 30 were subjected to the efficacy evaluation.

“For the efficacy endpoints, the median (interquartile range) was obtained for serum iPTH levels, and the mean (SD) was calculated for corrected Ca and P levels from week 28 to week 30 in each subgroup. The ratio of patients who achieved target levels of JSDT for iPTH (60–240 pg/mL), corrected Ca (8.4–10.0 mg/dL), and P (3.5–6.0 mg/dL) [19] were calculated from week 28 to week 30 using those who had levels above, within, and below the target ranges.”

---

## [Decision Letter · Decision Letter 2]

1 Dec 2022

Predictive factors requiring high-dose evocalcet in hemodialysis patients with secondary hyperparathyroidism

PONE-D-21-08923R2

Dear Dr. Tokumoto,

We’re pleased to inform you that your manuscript has been judged scientifically suitable for publication and will be formally accepted for publication once it meets all outstanding technical requirements.

Kind regards,

Gianpaolo Reboldi, MD, MSc, PhD

Academic Editor

PLOS ONE

Additional Editor Comments (optional):

Reviewers' comments:

Reviewer's Responses to Questions

**Comments to the Author**

1. If the authors have adequately addressed your comments raised in a previous round of review and you feel that this manuscript is now acceptable for publication, you may indicate that here to bypass the “Comments to the Author” section, enter your conflict of interest statement in the “Confidential to Editor” section, and submit your "Accept" recommendation.

Reviewer #2: All comments have been addressed

2. Is the manuscript technically sound, and do the data support the conclusions?

Reviewer #2: Yes

3. Has the statistical analysis been performed appropriately and rigorously? 

Reviewer #2: Yes

4. Have the authors made all data underlying the findings in their manuscript fully available?

Reviewer #2: No

5. Is the manuscript presented in an intelligible fashion and written in standard English?

Reviewer #2: Yes

6. Review Comments to the Author

Reviewer #2: (No Response)

7. PLOS authors have the option to publish the peer review history of their article (what does this mean?). If published, this will include your full peer review and any attached files.

Reviewer #2: No

---

## [Editor Report · Acceptance letter]

5 Dec 2022

PONE-D-21-08923R2 

Predictive factors requiring high-dose evocalcet in hemodialysis patients with secondary hyperparathyroidism 

Dear Dr. Tokumoto:

I'm pleased to inform you that your manuscript has been deemed suitable for publication in PLOS ONE. Congratulations! Your manuscript is now with our production department. 

Kind regards, 

on behalf of

Prof Gianpaolo Reboldi 

Academic Editor

PLOS ONE